# Federated Transformer-based Lightweight Modeling for Epilepsy Prediction using Twofold Personalization

George Anim
*Department of Computer Science and Engineering*
*Mississippi State University*
Mississippi, USA
gba37@msstate.edu

*Abstract*— **Automated epilepsy diagnosis research aims to improve prediction models using Electroencephalography (EEG) signals. Federated Learning (FL) preserves medical data privacy while accessing knowledge from multiple clients. Despite addressing data scarcity through collaboration, designing lightweight and personalized predictions with federated transformers for distributed EEG data is challenging. Additional modalities provide complementary knowledge for enhanced predictions. Developing a lightweight model for early, accurate personalized seizure prediction from multimodal signals offers significant opportunities. This work introduces client-level and patient-specific personalization using a federated transformer model. The self-attention mechanism in federated transformers can negatively impact results due to data heterogeneity, limiting collaboration. Hypernetwork addresses this by learning personalized self-attention layers, generating EEG-representative attention maps, and eliminating global self-attention aggregation. Model parameters are aggregated globally. Client-level personalization uses a local transformer (teacher model) for prediction. Knowledge distillation creates a lightweight patient-level model (student model) from teacher model weights, integrating multimodal signals for patient-specific prediction. Validated with the MIT-CHB dataset, this approach accurately determines the preictal state, outperforming existing models proved by potential outcome of 95.58% sensitivity and 98.61% specificity. Also, the proposed approach yielded only a 0.014 False Positive Rate (FPR) while finetuning the student model of each hospital with the multimodal data by the federated-guided generalized knowledge from the teacher model.**

*Keywords*— *Client-level, Epilepsy Prediction, Federated Transformer, Hyper network, Knowledge Distillation, Personalization, Multimodal, Patient-Specific Personalization, and TwoFold Personalization*

## I. INTRODUCTION

Epilepsy [1] is a neurological disorder characterized by recurrent, unprovoked seizures, which are sudden, abnormal electrical discharges in the brain. The apparent unpredictability of epilepsy, such as the uncertainty in the seizure occurrence for each patient, affects the quality of life for individuals. During an epileptic seizure [2], a substantial quantity of brain neurons participates in an excessive, synchronized, and inappropriate electrical discharge, leading to the manifestation of signs and symptoms. The epileptic seizure detection and prediction profoundly influence the daily lives of epileptic patients. Providing precise last-minute notifications about impending seizures promptly leverages safety precautions [3]. In epilepsy diagnosis, scalp or intracranial EEG signals enable the exploration of different transitions of the epileptic seizure states. In epileptic EEG signals, four stages of epileptic seizures involve the interictal, preictal, ictal, and postictal. Epilepsy prediction aims to predict pre-seizures that are preictal states, focusing primarily on two phases: the interictal phase, representing the interval between two epileptic seizures during which the patient exhibits apparent normal behavior, and the preictal phase, characterized by the abrupt onset of sporadic isolated spikes and large amplitudes [4, 5]. The epilepsy landscape has transformed due to recent progress in data-driven computational methods and device technology, turning the accurate prediction of seizures into a tangible reality. Deep learning (DL) stands out by accurately learning patterns from extensive data through the classification process within intricate hierarchical structures. In personalized healthcare, continuous remote monitoring, and emergency intervention, the Internet of Things (IoT) emerges as a remarkable technology. IoT establishes connections among various devices or components as middleware, facilitating seamless communication for predicting and monitoring seizures [6].

Recent research shows promising results in probabilistic seizure risk prediction using wearable devices and electronic diaries. However, effective prediction relies on continuous monitoring, EEG signals, and the challenge of developing a lightweight, privacy-aware design for remote data integration [7]. Personalizing seizure prediction faces data scarcity and inter-patient variability due to limited EEG patterns within hospitals [8]. Developing personalized models with available patient data, privacy, and computational efficiency is challenging with single-modality DL models like EEG. To address these issues, research has used FL [9] for data scarcity and privacy and transformer models [10] for learning relationships in epileptic inputs.

Multimodal inputs from wearable and implantable devices enhance seizure prediction, providing continuous EEG data and information on motor behavior, electrocardiogram (ECG), electromyography (EMG), accelerometry (ACC), and photoplethysmography (PPG) data from the autonomic nervous system. This makes epilepsy prediction more effective. However, designing a lightweight system that combines FL and transformer models without compromising accuracy is challenging. Therefore, developing a personalized, lightweight

epilepsy prediction model is essential for improving the quality of life for epileptic individuals through collaborative training, privacy preservation, and model customization.

The significant contributions of this work are presented as follows.

- This work presents a twofold personalization for epilepsy prediction: client-level and patient-specific personalization.
- Fold1 offers client-level personalization while applying the federated transformer model with the adoption of hypernetwork for precisely understanding the client-level epileptic EEG pattern, which works as a teacher model.
- To enhance the learning ability of the transformer model in both the time and frequency domain, the proposed approach provides the input epileptic EEG data as the ternary feature representations involving the timestep, channel, and spectral embeddings.
- The client-level personalized attention maps are generated for each hospital with the help of EEG embedding input-based hypernetwork modeling. At the same time, the self-attention layer is retained in the local model while the model parameters are aggregated with the global model.
- Fold2 is a patient-specific personalization that applies the knowledge distillation from the client-level personalized teacher model and utilizes multimodal inputs, such as the ECG, PPG, and ACC, acquired from the patients' wearables to ensure the lightweight prediction.
- Thus, the proposed approach precisely identifies the preictal state and predicts the seizure onset in the sequence of epileptic EEG time series data.

## II. LITERATURE REVIEW

This section investigates recent research developments through a comprehensive literature review of various epileptic seizure prediction approaches with DL and advanced transformer architectures.

### 2.1. Epilepsy Prediction Approaches

To predict epileptic seizures, patient physiological signals are acquired invasively or noninvasively. Several ML algorithms are vital for epilepsy prediction. For instance, decision trees in [11] predict seizures using electronic seizure diaries that record mood, symptoms, stress, and seizure occurrences. The approach in [12] uses XGBoost on 36-minute pre-seizure data to distinguish preictal from interictal states, optimizing the interval with Leave-One-Patient-Out cross-validation. In [13], the Multiscale Prototypical Part Network (MSPPNet) DL model captures multiscale EEG features for enhanced decision-making.

Research [14] presents a patient-specific model combining a sparse autoencoder and SVM classifier to categorize EEG signals. Study [15] uses a long short-term memory (LSTM) RNN with a wrist-worn sensor for real-world seizure forecasting, outperforming a random predictor in most cases. The approach in [16] employs an LSTM model on subcutaneous EEG recordings for patient-specific predictions.

ForeSeiz [17] integrates an ECNN and a Phase Transition Predictor for real-time seizure prediction. Research [18] transforms EEG data into temporal and spectral features, using Principal Component Analysis (PCA), Common Spatial Pattern (CSP), and Multivariate Multiscale Sample Entropy (MMSE) for temporal analysis and unified Maximum Mean Discrepancy Autoencoder (uMMD-AE) for spectral analysis, followed by SVM decision fusion. The two-layer LSTM model in [19] uses spectral features to distinguish preictal and interictal states.

DL models face challenges like interpretability and computational efficiency in epilepsy prediction due to evolving EEG patterns and data scarcity. Consequently, FL and transformer models are increasingly used for accurate epilepsy detection and prediction, maintaining data privacy even in data-scarce environments.

### 2.2. Federated Learning and transformer-based Epilepsy Prediction Approaches

Personalized FL framework [20] explores Convolutional Neural Network (CNN) architectures for EEG-based seizure detection, balancing performance and energy consumption. It allows clients to use personalized DNNs collaboratively, minimizing communication energy. Decentralized FL [21] employs adaptive ensemble learning for seizure detection, addressing non-independent and identically distributed (non-IID) challenges within hospitals and aligning with wearable system constraints.

Transformer architectures parallelize input sequences, reducing training time and enhancing interpretability for disease diagnosis. EpilepsyNet [22] uses transformer models for seizure detection from EEG signals, applying Pearson Correlation Coefficient (PCC) to compute statistical relationships between features. The approach in [23] emloys Sequence Transformer Network (STN) which learns temporal changes in EEG data, predicting seizures with a CNN model using Short-Time Fourier Transform (STFT) transformed features. A three-tower transformer model [24] fuses time and frequency domain features for seizure prediction. The prediction model in [25] uses Temporal Multichannel Transformer (TMC-T) and Vision Transformer (TMC-ViT) for multichannel EEG signals, examining seizure prediction performance with varying sample sizes.

## III. PRELIMINARIES, PROBLEM FORMULATION AND SYSTEM MODEL

This section introduces the preliminary information for developing the epilepsy prediction approach, problem formulation, and system model.

### i) Preliminaries

Federated Learning: FL allows organizations to train models without sharing private data. Hospitals collaborate, training models on diverse patient data without exchanging specifics. FL improves generalization by utilizing data from multiple sources, maintaining the same distribution or feature space but

varying sample spaces.

Transformer: Transformers capture long-range dependencies in EEG data, crucial for identifying epileptic activity patterns. They automatically learn representations from EEG data, focusing on essential segments and reducing noise through self-attention.

Hypernetwork: Hypernetworks produce parameters for other networks, dynamically generating features from input data. They adapt model parameters based on input data properties, serving as regularization for transformer-based epilepsy prediction.

Multimodality: Utilizing various information sources improves prediction accuracy by capturing a broader range of parameters contributing to epileptic activity. Integrating data from multiple physiological signals results in highly robust predictions through a combination of rich and informative feature representations.

Knowledge Distillation: It constructs efficient prediction models for wearables by training the student model to imitate the teacher model's predictions. The student model minimizes distillation loss, transferring knowledge from the teacher model, suitable for resource-constrained devices.

TABLE I. MODELS AND THEIR SIGNIFICANCE IN THE PROPOSED SYSTEM

| Constraints in Epilepsy Prediction | Factors to Resolve Constraints | Concept | Generic Advantages | Epilepsy-specific Advantages | Twofold Design of This Work |
|---|---|---|---|---|---|
| Data Scarcity and Data Privacy | Federated Learning | Collaborative training without sharing raw data | Diversity Privacy Edge Computing | Patient Privacy Personalization Diversified representation learning | Fold 1 |
| Data Interpretability and Prediction Accuracy | Transformer | Context-aware representation learning with self-attention and position encoding | Parallelization Long-range dependencies | Efficient training of EEG recordings Long-range dependency in epilepsy state analysis | Fold 1 |
| Lack of Personalization | Hyper network | Parameter generation | Efficiency in parameter Regularization | Interpretability in seizure states Adaptable to personalized EEG patterns | Fold 1 |
| Data Scarcity and Lack of Personalization | Multimodality | Integration of multiple inputs | Complementary knowledge Inherent understanding | Comprehensive patient-specific seizure knowledge Scarcity Handling | Fold 2 |
| Need for Lightweight Design | Knowledge Distillation | Knowledge transfer from teacher to student model | Computation efficiency | Model compression Personalization | Fold 2 |

## ii) Problem Formulation and System Model

The epileptic seizure is still an open research problem, and accomplishing high sensitivity and specificity is challenging. The seizure prediction algorithms rely on the patient intervention or wearables implanted in the patient's body, employing various physiological signals. Epilepsy prediction is formulated as the binary classification problem, discrimination of preictal and interictal state. To recognize epileptic seizure activities, the widely applied physiological signals involve the EEG, ECG, PPG, and ACC, acquired from the electrodes and wearable devices.

This work designs a twofold personalization-based epilepsy prediction model using federated transformer learning with personalization and knowledge distillation. In Fold1, the proposed system employs the FL model for healthcare services among multiple distributed hospitals, particularly EEG-based epilepsy prediction. For the 'N' number of hospitals, the proposed approach utilizes the 'N' number of different epileptic EEG datasets, and the 'N' number of similar transformer models accompanied by a multi-head self-attention mechanism. The generalization relies on the epileptic EEG signals, and the personalization relies on additional physiological signals. To implement the federated scenario for the 'N' number of hospitals with distinct epileptic EEG data distribution, applying the federated aggregation is challenging.

$$arg \min_{\varphi} \sum_{i=1}^{N} \frac{s_i}{S} \mathcal{L}_i(\eta_i) \qquad (1)$$

As formulation in (1), the proposed epilepsy prediction aims to minimize the personalized loss of each local model or client (i) while aggregating each client's local EEG samples ($s_i$) in a unified manner (S). $\varphi$ and $\eta_i$ refers to a set of personalized parameters and a personalized local model. Moreover, $\mathcal{L}_i$ denotes the loss function of patient samples in each client, referring to the cross-entropy loss. To minimize the personalized loss in the federated model, the integration of FL with the transformer model is enhanced by the hypernetwork-based personalization performed during the local model updation at the end of the FL-based prediction task over the iterations. In particular, the federated aggregation in the proposed system combines the partially transferred local model parameters to maintain data privacy while applying the self-attention-based transformer model. The main objective of Fold2 design is to accurately predict epilepsy by discriminating the personalized preictal and interictal states from the EEG signals through knowledge distillation with the additional knowledge of other physiological signals, such as the ECG, PPG and ACC. In the task of knowledge distillation, optimizing the personalization of each local model based on its personalized inputs is challenging without compromising the personalization of the local model and the patients. Therefore, the knowledge distillation in the proposed epilepsy prediction

is based on the minimization of prediction loss in each client, $\mathcal{L}_i(\eta_i)$ as well as each patient, $\mathcal{L}_i(\mu_i)$.

$$arg \min_{\omega} \mathcal{L}_i(\omega_i) = \mathcal{L}_i(\eta_i) + \gamma.\mathcal{L}_i(\mu_i) \qquad (2)$$

As mentioned in (2), loss of prediction personalization with the distance measure in knowledge distillation, $\mathcal{L}_i(\omega_i)$ is the cumulative measure of client-level and patient-level personalization; hence, the proposed system transfers the client-level personalized knowledge as the global representation for multimodal-associated patient-level personalization. In each hospital, the multimodal inputs are gathered from the wearables of corresponding patients enrolled in the hospitals or clients for enhancing epilepsy prediction.

## IV. SEIZURE PREDICTION METHODOLOGY

The proposed model designs a twofold personalization involving the federated transformer learning-based client-level personalization, considered a teacher model and the knowledge-distillation-based patient-specific personalization a student model. In the proposed system, Fold1 intends to integrate the FL with the transformer model to utilize the global epileptic patient knowledge collaboratively. However, the inherent heterogeneity in the hospitals' data makes generalizing the globally shared parameters-based local seizure prediction at the edge ineffective. Hence, the federated transformer model integration is based on the hypernetwork built by embedding EEG input for the client-level personalization in the self-attention mechanism. Moreover, Fold 2 employs knowledge distillation from the teacher model for patient-specific personalization by utilizing multimodal data, including ECG, PPG, and ACC, captured from wearable devices to ensure the lightweight design of seizure prediction. Figure 1 shows the proposed federated transformer model for epilepsy prediction.

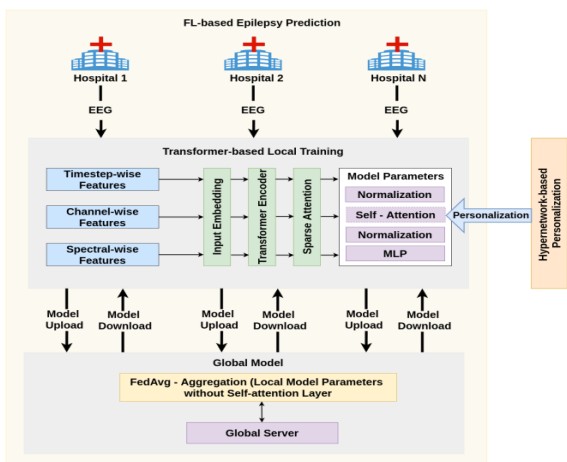

Figure 1: Federated Transformer Model in Proposed Epilepsy Prediction

### 4.1. Data Description and Preprocessing

The proposed system uses the CHB-MIT scalp EEG dataset [26] with recordings from 24 patients at Boston Children's Hospital. EEG signals, sampled at 256 Hz, include 198 seizures. Preictal data is 1 hour before a seizure, interictal data spans 4 hours around seizures. Utilizing EEG data from 21

patients (excluding CHB12, CHB13, and CHB24), the system predicts 108 seizures. Besides EEG, it incorporates physiological signals from wearable devices like ECG, PPG, and ACC, captured by ePatch [27] for ECG and Empatica E4 [28] for PPG and ACC. Signals undergo bandpass filtering: EEG (0.5-40 Hz), ECG (0.4-20 Hz), PPG (0.4-3 Hz), and ACC (0.5-40 Hz). ECG signals are segmented into 60s epochs and notch filtered to remove 50 Hz interference.

For time series analysis, STFT transforms EEG signals into 2D matrices for time-frequency domain analysis. Z-score normalization is applied to EEG channels within each patient's samples. Each EEG segment is represented as an N×M matrix, with N as the 256-segment duration and M as the 23 channels in the CHB-MIT dataset, providing a generic representation for all patients in epilepsy classes.

### 4.2. Fold1: Client-level Personalization

The client-level personalization uses a federated transformer network for EEG data, ensuring data privacy and improving generalization. It offers personalized seizure prediction by learning personalized self-attention layers to address data heterogeneity. The global model aggregates local parameters without the self-attention layer, while local execution uses hypernetwork-assisted attention maps. Epileptic EEG input is represented with timesteps, channels, and spectral features in a transformer model with multi-head self-attention. Positional encoding is applied to timestep and spectral features but not to channel-wise features. Spectral features like delta, theta, alpha, beta, and gamma bands highlight brain variations for seizure prediction. Moreover, the distinct features are extracted from the spectral sub-bands to characterize the epileptic seizure for prediction [29].

To ensure a lightweight model, sparse attention [30] reduces computation complexity from $O(n^2)$ to $O(n \log n)$, enabling deployment on resource-constrained devices. Generalized self-attention in FedAvg can degrade performance, leading to convergence and scalability issues. During local training, ternary features are learned using normalization, self-attention, and MLP layers to recognize the preictal state. Only partial transformer model parameters are uploaded to the global model, retaining local self-attention layers to preserve patient-specific patterns. The global model aggregates local parameters to recognize the preictal state using knowledge from multiple hospitals.

Hypernetwork-based Personalization: FedAvg aggregation of self-attention layers affects performance with heterogeneous data. Hence, this approach uses personalized self-attention presented in [31][32] for EEG data at the client level. The hypernetwork fine-tunes the self-attention layer and captures global model parameters without compromising privacy. It generates attention maps for client-level personalization while applying the FL model. For EEG input, the hypernetwork generates projection matrices to fine-tune the self-attention layer in each hospital's personalized epilepsy prediction. It learns the embedding vector of EEG features and generates

partial weights for each client's seizure prediction model. This generates attention maps for each client, improving client-level seizure prediction in a distributed environment.

Client-Level Personalization: Client model updating involves global model parameters and hypernetwork-based personalized EEG data representation. Jointly training the local model with global parameters and hypernetwork-generated attention maps improves seizure prediction accuracy. This federated transformer model enhances EEG-based seizure prediction by accurately discriminating between epilepsy states for each client.

### 4.3. Fold2: Knowledge-distillation-assisted Patient-Specific Personalization

Fold2 personalization is a lightweight, patient-specific approach using knowledge distillation from the generalized federated transformer model in Fold1. This teacher-student paradigm builds a lightweight student model from the teacher model's knowledge. The system fine-tunes client-level predictions based on global knowledge distance measures. Instead of using large-scale EEG data, it utilizes multimodal physiological data, enhancing computational efficiency. The combined knowledge from the federated transformer model and multimodal inputs in the student model accurately identifies the preictal state. For seizure patients, wearable ECG devices like ePatch record ECG signals, aiding seizure detection through HRV analysis and R-peak detection. Combining ECG and EEG improves prediction accuracy by measuring responses from both the central and peripheral nervous systems. Changes in HRV, indicating autonomic nervous system activity, help predict seizures.

The Empatica E4 device monitors PPG and ACC signals for epilepsy. PPG measures blood volume changes, while ACC detects movement patterns, both aiding in seizure prediction. Integrating EEG, ECG, PPG, and ACC data improves the model's specificity and sensitivity. The Pan-Tomkins algorithm determines R-peaks in ECG signals, computing HRV from these peaks. Pulse Rate Variability (PRV) and Pulse Transit Time (PTT) from PPG signals further aid prediction, with PTT indicating blood pressure changes linked to seizures.

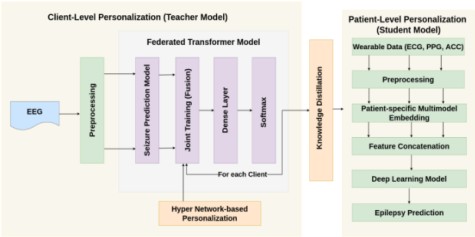

Figure 2 Knowledge Distillation Process for Patient-Specific Personalization

As shown in Figure 2, the proposed approach uses knowledge distillation from the teacher model, which contains client-level personalized knowledge, to personalize the student model for patients. Instead of using knowledge from the global model, seizure prediction distills knowledge between the hospital and

patient. This iterative finetuning reduces computation complexity and improves accuracy. Heterogeneity among hospitals affects the student model's prediction quality when using generalized global knowledge. Hypernetwork-based attention maps in client-level personalization compute the distance between knowledge representations, regulating the student model with minimal computation for each patient.

Wearable technologies assist in dynamic human analysis by monitoring disease evolution over time. They provide contextual data for personalized disease monitoring and allow clinicians to respond to physiological changes in real-time. The proposed approach uses multimodal data from wearables for patient-specific epilepsy personalization, identifying the preictal state. Figure 3 shows patient-specific personalization based on multimodal data.

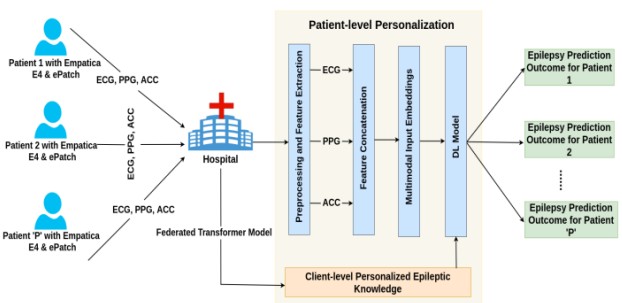

Figure 3: Multimodal-based Patient-Specific Personalization

In Algorithm 1, the detailed procedure of the proposed epilepsy prediction is presented as follows.

**Algorithm 1: Pseudocode of TwoFold**
**Input:** EEG, ECG, PPG, and ACC
**Output:** Seizure State Prediction via Preictal identification
**//Fold 1 - Client-level Personalization//**
**1 for** all the clients /hospitals **do**
**//Preprocessing and Feature Extraction//**
**2**    **for** all the EEG samples in each hospital **do**
**3**      Apply filtering and normalization
**4**      Extract ternary features (timestep, spectral, and channel)
**5**    **endfor**
**//Federated Transformer-based Learning//**
**6**    **for** all the extracted EEG features in all the hospitals **do**
**7**      Apply federated learning
**8**      Execute local model individually by the transformer model
**9**      Apply the sparse attention for lightweight execution
**10**      **for** each local model **do**
**11**        Personalize the self-attention by the hypernetwork
**12**        **for** each local epileptic EEG samples in each hospital **do**
**13**          Generate epileptic EEG embedding input for the hypernetwork
**14**          Generate the attention maps for transformer model
**15**        **endfor**
**16**      **endfor**

17    Train the local model for each hospitals' epileptic EEG data

18    Transfer the local model parameters for the aggregation without self-attention

**//Collaborative Federated Training//**

19    Perform federated aggregation and global model training

20    **if** global model execution reach epochs **then**

21        Transfer global model parameters to each local model

22    **endif**

**//Personalization//**

23    **for** each local model or client **do**

24        Perform joint training of global model and hypernetwork-based parameters

25        Update local model with personalized self-attention in each hospital

26        Predict epilepsy by the classification of preictal and interictal

27    **endfor**

28    **endfor**

**//Fold 2 – Patient-level Personalization//**

30 **for** each hospital **do**

31    Perform knowledge disillation

32    Apply the deep learning model

33    **for** each patient **do**

34        Acquire multimodal physiological signals from each patient

35        **if** distance between epileptic patterns teacher and student model is minimal **then**

36            Transfer client-level personalized knowledge to the student model

37        **endif**

38    **endfor**

39    Predict the epileptic seizure via preictal identification

40 **endfof**

## V. EXPERIMENTAL EVALUATION

The experimental framework implements the epilepsy prediction algorithm using Python programming, assessing the prediction performance by classifying pre and interictal classes. The experimental model utilizes the EEG, ECG, PPG, and ACC data for the proposed model implementation. EEG data is utilized from the CHB-MIT dataset, and ECG, PPG, and ACC data are assumed as observed from the corresponding patients' wearables. The simulation of a multimodal scenario in the proposed epilepsy prediction system is conducted to demonstrate the proposed epilepsy prediction prototype. The existing epilepsy prediction research [24, 25] is evaluated under the parameter settings mentioned in the corresponding research work for the CHB-MIT EEG dataset. The proposed seizure prediction model is trained and tested on the client-level and patient-level personalization stages using cross-validation. The evaluation of baseline DL models is implemented for the same set of CHB-MIT EEG datasets with the default learning parameters with the train-test split validation. To assess the performance of the epilepsy prediction, the experimental model employs precision, recall or sensitivity, specificity, False Positive Rate (FPR), and Area Under the ROC Curve (AUC).

**Precision:** Ratio between the number of correctly detected preictal samples and the number of detected samples in the preictal state. **Sensitivity:** Ratio between the number of correctly detected preictal samples and the total number of actual samples in the preictal state.

**Specificity:** Ratio between the number of correctly detected interictal samples and total number of actual samples in interictal state.

**False Positive Rate:** Ratio between the number of incorrectly detected preictal samples and total number of interictal samples.

**AUC:** Measure classification performance in assessing the degree of discrimination between pre and interictal classes.

TABLE II.    MODEL PARAMETERS

| Parameters | Values | |
|---|---|---|
| | Teacher Model (Federated Transformer) | Student Model |
| Number of Clients | 3 | 3 |
| Number of Communication Rounds | 3 | - |
| Dropout Rate | 0.1 | 0.2 |
| Learning Rate | 0.01 | 0.01 |
| Activation | RELU, Sigmoid | RELU, Sigmoid |
| Loss Function | Binary Cross Entropy | Binary Cross Entropy |
| Epochs | 10 | 5 |
| Batch Size | 32 | 2 |
| Optimizer | Adam | Adam |

### 5.1. RESULTS AND DISCUSSION

The performance of the proposed epilepsy seizure prediction is compared with several baseline algorithms and two existing epileptic seizure prediction research works [24, 25].

TABLE III.    COMPARATIVE ELEMENTS IN PROPOSED EPILEPSY PREDICTION SYSTEM AND EXISTING SYSTEMS

| Epilepsy Prediction Elements | | | | | | | |
|---|---|---|---|---|---|---|---|
| Epilepsy Prediction | Deal with Data Scarcity | Data Privacy | Data Interpretability Analysis | Parallelization | Personalization | Multi modality | Light Weight |
| [24] | ✕ | ✕ | ✕ | ✓ | ✕ | ✕ | ✕ |
| [25] | ✕ | ✕ | ✕ | ✓ | ✕ | ✕ | ✕ |
| Proposed | ✓ | ✓ | ✓ | ✓ | ✓ | ✓ | ✓ |

Note: ✓ - Presence, ✕ - Absence

Table 3 compares the proposed epilepsy prediction system with two existing epilepsy prediction systems regarding research constraints. The proposed epilepsy prediction performance is provided in Table 4, while the prediction outcome is assessed two-fold for the CHB-MIT epileptic EEG dataset and multimodal physiological signals.

TABLE IV. TWOFOLD PERSONALIZATION PERFORMANCE OF THE PROPOSED EPILEPSY PREDICTION

| Approach | Performance of Epilepsy Prediction | | | | | |
|---|---|---|---|---|---|---|
| | Client-level Personalization (Fold 1) | | | Patient-Level Personalization (Fold 2) | | |
| | Sensitivity (%) | Specificity (%) | FPR | Sensitivity (%) | Specificity (%) | FPR |
| Proposed | 87.04 | 78.71 | 0.209 | 95.58 | 98.61 | 0.014 |

From Table 4's result analysis, the proposed seizure prediction method in Fold 2 achieved higher true positives and negatives than client-level personalization in Fold 1. Integrating multimodal physiological signals significantly impacts prediction outcomes using distilled knowledge from client-level personalized EEG representations. Lower FPR values in Tables 4 and 5 indicate better performance in accurately identifying the preictal state, showcasing the proposed system's optimal performance despite scarce epileptic knowledge. FL addresses scarcity in Fold 1 and multimodality in Fold 2, resulting in higher sensitivity, specificity, and minimal FPR.

Table 4 compares epilepsy prediction performance at the patient level with client-level personalization, showing patient-level sensitivity and specificity are 8.54% and 19.9% higher due to multimodal knowledge-based finetuning. Table 5 shows the proposed model outperforms baseline models and existing research (TMC-T [24] and three-tower transformer model [25]) across all measures, including precision, sensitivity, specificity, AUC, and FPR. The proposed method achieves a 95.58% true positive rate and 98.61% true negative rate, 0.11% and 3.49% higher than the comparative model [25]. Scarce personalized and diversified EEG knowledge affects the performance of the comparative model for new samples.

TABLE V. COMPARISON OF THE PROPOSED EPILEPSY PREDICTION WITH THE BASELINE MODELS AND EXISTING EPILEPSY PREDICTION WORKS

| Models | Average Epilepsy Prediction Performance | | | | |
|---|---|---|---|---|---|
| | Precision (%) | Sensitivity (%) | Specificity (%) | AUC (%) | FPR |
| Random Forest | 86.68 | 88.37 | 86.31 | 87.34 | 0.13 |
| 1DCNN | 85.18 | 84.14 | 87.75 | 85.95 | 0.12 |
| 2DCNN | 76.31 | 70.73 | 81.63 | 76.18 | 0.18 |
| TMC-T [25] | 86.21 | 92.59 | 85.18 | 88.88 | 0.14 |
| Three-tower transformer [24] | 94.89 | 95.47 | 95.12 | 95.41 | 0.048 |
| Proposed | 98.59 | 95.58 | 98.61 | 98.02 | 0.014 |

Table 5 compares the proposed seizure prediction model with baseline models [24, 25] using CHB-MIT EEG samples. The proposed model combines federated learning and the transformer model, benefiting from collaborative training of diverse epileptic EEG patterns and efficient parallel execution. It outperforms the TMC-T model, with a 2.99% higher sensitivity for preictal state identification and improved FPR. The true positive rate indicates accurate seizure prediction by distinguishing preictal from interictal states. The proposed model achieves 98.61% specificity and 97.09% AUC, outperforming baseline models. Figure 4 shows the superior performance of the proposed model compared to existing works [24, 25].

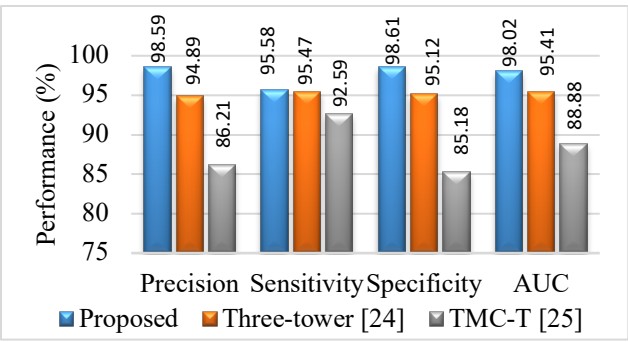

Figure 4: Comparison of Epilepsy Prediction Performance

The AUC and accuracy performance of the proposed seizure prediction approach are plotted in Figure 4 for the tested outcome of the CHB-MIT EEG dataset. In order to validate the classification model performance, the experimental model assesses the accuracy and AUC that measures the trade-off between the sensitivity and specificity values. In particular, AUC measures the class discrimination ability of the learning model for the binary classes of preictal and interictal. The teacher model-based student model finetuning facilitates the accurate detection of classes from learning multimodal input patterns in the proposed approach, resulting in 98.02% and 95.58% AUC and sensitivity, respectively. Compared to the three-tower transformer model, the proposed approach obtained 2.61% higher AUC due to the potential advantage of learning the client-level personalized knowledge from the diversified EEG patterns.

## VI. CONCLUSION

A twofold personalization-based epilepsy prediction model is proposed based on client-level and patient-specific personalization to enhance the epileptic seizure prediction. The client-level personalization is considered a teacher model, and the patient-specific personalization is a student model. Moreover, the hypernetwork, multimodality, and transformer

models resolve the personalization and prediction accuracy limitations. The hypernetwork-based personalization aided in retaining the contextual epileptic patterns learned by the transformer in each hospital, and federated aggregation provided the globally learned diversified epileptic EEG knowledge from different hospitals, thereby improving the epilepsy prediction performance at the local level. Client-level personalization is achieved by joint learning between the hyperparameter-enabled self-attention and extracting the knowledge from the federated transformer model from the epileptic EEG data. The patient-level personalization is facilitated in fold 2 for each client while utilizing the multimodal inputs observed from the patients' wearables. Thus, the proposed approach effectively discriminates the preictal and interictal state of the epileptic samples in the student model from the influence of the teacher model.

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
