# OpenReview forum: "Federated Transformer-based Lightweight Modeling for Epilepsy Prediction using Twofold Personalization"
_IEEE.org/EMBS/BHI/2024/Conference — IEEE BHI'24_

### Official Review · Reviewer_YpDA · 2024-08-08
**The study introduced a comprehensive pipeline to predict epilepsy and address several existing challenges.**

**Overall Rating:** 7
**Confidence:** 3

**Other Quality Metrics:**

(a) Clarity of writing: Fair

(b) Clinical Significance: Good

(c) Methodological Novelty: Great

(d) Experiments and Results: Great

**Questions For The Authors:**

1. The experiment is primarily based on the MIT-CHB dataset. Have you tried another dataset? Additional datasets could strengthen the generalizability of the results.

2. The study considers EEG, ECG, PPG, and ACC as important features for epileptic seizure prediction. Are there any other confounding variables related to epilepsy, and is it possible to incorporate them into your model?

**Strengths:**

1. Using federated learning ensures data privacy by training models across multiple clients without sharing raw data.
2. The knowledge distillation technique results in a lightweight model, reducing computation complexity.
2. The proposed model outperforms existing methods.

**Summary Of The Paper:**

This paper proposes a federated transformer-based model for epilepsy prediction to enhance early seizure detection from EEG signals while maintaining data privacy. The approach integrates client-level and patient-specific personalization to improve the prediction and computational performance.

**Weaknesses:**

1. The model's complexity and the integration of multiple techniques may pose challenges for practical implementation and also hard to interpret the model.
2.  The study only validates one dataset, lacking generalization.

---

### Official Review · Reviewer_QnVd · 2024-08-09
**Outside my domain but overall comprehensive and well written paper.**

**Overall Rating:** 7
**Confidence:** 2

**Other Quality Metrics:**

● Clarity of writing: Excellent
● Clinical Significance: Excellent
● Methodological Novelty: Excellent
● Experiments and Results: Excellent

**Questions For The Authors:**

NA

**Strengths:**

Comprehensively written paper. Presents the past works well and highlights the key benefits of the proposed approach. Then sufficiently demonstrated these benefits compared to existing works in the results section.

**Summary Of The Paper:**

Presents a structure for federated learning with personalisation for epilepsy prediction.

**Weaknesses:**

•	Good to provide some summary results in the abstract
•	It is mentioned that the aim is for a lightweight model. Would be good to highlight the size of the model and the processing time
•	Figure 4 is currently mentioned in-text as Figure 6. It appears to be unnecessary as AUC and accuracy are presented in Table 5 (unless I misunderstood). ROC curves on the same plot for the proposed and [24] and [25] models would be more beneficial as a figure.

---

### Decision · Program_Chairs · 2024-09-23

Accept